

# Analysis of measurement differences and causes of C, N, and P in river flooding areas—taking the Hailar River in China as an example

Xi Dong and Chunming Hu

National Engineering Research Center of Industrial Wastewater Detoxication and Resource Recovery, Research Center for Eco-Environmental Sciences, Chinese Academy of Sciences, Beijing, China

## ABSTRACT

The Hailar River is located in the Inner Mongolia Autonomous Region of Northeast China. It is a connecting hub of the agricultural pastoral transitional zone on the Hulunbuir grassland, with abundant water and biodiversity resources, and important ecological conservation significance. This study takes the Hailar River as the research area to evaluate the impact and main influencing factors of soil C, N, and P ecological measurement from the upstream to downstream concave convex riverbanks of the Hailar River. The research results show that: (1) The average soil particle size shows differences in the upstream and downstream: the average soil particle size in Section 1 is 31.6–192.3 µm, Section 2 is 21–213 µm, Section 3 is 21–288 µm, and Section 4 is 42–206 µm; the pH value in the upstream area is generally low, while the pH value in the downstream area increases. The reason for this is that the convex bank area has sufficient water, which plays a role in inhibiting salt content; (2) the nutrient content in the concave bank is generally higher in the upstream region than in the downstream region, while the difference in nutrient content between the upstream and downstream regions is relatively small in the convex bank; (3) the nutrient content of concave banks is mostly positively correlated with soil moisture content, while convex banks are positively and negatively correlated with soil moisture content and soil particle size. Research has shown that different cross-sections upstream and downstream, as well as uneven riverbanks, significantly affect soil physicochemical properties and soil C, N, and P ecological measurements. Studying the content of soil C, N, and P in different riparian zones under typical cross-sections can provide new ideas for regional ecological protection and even global C, N, and P cycling.

# INTRODUCTION

The Hailar River Basin is located in the northernmost part of the Inner Mongolia Autonomous Region. The Hailar River riparian zone is a transitional and connecting zone between the Hulunbuir Grassland and rivers or lakes, with rich ecosystem types, water and feed resources (*Zhu et al., 2019*; *Zhu et al., 2018*). At the same time, the riparian

Corresponding author
Chunming Hu, cmhu@rcees.ac.cn

zone and its surrounding floodplain wetlands are also important habitats for migratory birds and wildlife (*Li et al., 2020*). The Hailar River is an important ecological protection barrier and conservation area in Northeast China, which has a significant impact on maintaining regional ecosystem stability, maintaining ecosystem diversity, and promoting regional economic development.

Due to recent global climate change and the impact of global warming, the natural conditions in the Hulunbuir region have also been disrupted. For example, changes in vegetation cover caused by climate change in the Hailar River grassland, as well as changes in biomass over the past decade (*Dong & Hu, 2021*; *Dong & Hu, 2022*; *Abbas et al., 2023*; *Kayitesi, Guzha & Mariethoz, 2022*). At the same time, the development of local society and economy, the construction of water conservancy hubs, and other comprehensive factors have led to a decrease in the level of the Hailar River, The area near the edge of the riparian zone is greatly affected by climate change and the alternating cycles of river inundation and drought (*Wu & Liu, 2015*), which will greatly affect the soil physical and chemical properties of plant diversity, soil nutrient deposition, and soil C, N, and P content in the riparian zone (*Jiang et al., 2015*; *Li et al., 2020*; *Wilson et al., 2011*).

Rivers are an important natural force in shaping grassland terrain, and particulate matter and nutrients are transported upstream and downstream along the river basin. At the same time, with the erosion and sedimentation of the river, the river shoreline constantly fluctuates and changes, forming differentiated ecosystems on concave and convex banks, bringing different carbon emissions, ecological services, and so on. For riparian ecosystems, C, N, P, and other nutrients in soil sediment play the most important role in the Earth's material cycle and balance mechanisms (*Sinsabaugh & Follstad Shah, 2012*). Soil nutrients are involved in important soil biological processes, such as the coupling and transformation of C, N, and P (*Lan et al., 2022*; *Li et al., 2018*), soil litter decomposition, as well as nutrient and organic matter decomposition or participation in the Earth's material cycle (*Li et al., 2013*). Meanwhile, riparian sediments C, N, and P can be used as important indicators for evaluating the health quality of riparian zones (*Huang, Wang & Ren, 2019*). Previous studies have mainly focused on exploring changes in the frequency of riverbank inundation and its impact on plant succession in the riparian zone (*Dong & Hu, 2021*; *Dong & Hu, 2022*), and research has focused on agricultural soil, such as agricultural soil erosion, physical and chemical characteristics, greenhouse gas emissions (*Guo & Jiang, 2019*; *Jacinthe et al., 2015*). Some scholars have also focused on plateau grassland C, N Research on the P-cycle (*Lu et al., 2023*) and spatial refinement of C, N, and P in desert soils (*Lan et al., 2022*; *Zhang, Su & Yang, 2019*). However, there is limited research on soil C, N, and P in the Hailar River riparian zone, especially in typical cross-sections where there is insufficient research on the ecological measurement of C, N, and P in different riparian zones. Therefore, this study is conducted from the perspective of soil nutrient content under typical sections of the riverbank zone, in order to provide support for the study of ecological system development and changes on both sides of the river.

In response to the existing problems in the above research, this study takes the Hailar River in Inner Mongolia Autonomous Region, China as the research object, and explores

the impact of different riverbank zones (concave and convex banks) from the upstream to downstream of the Hailar River on the ecological measurement of soil TC, AHN, and AP, as well as the scientific issues of limiting factors. This study aims to achieve the following three main objectives: (1) Quantitatively evaluate the differences in physical and chemical properties and nutrients of soil on concave and convex banks; (2) revealing the changes and correlation between the physicochemical properties and nutrients of the concave and convex banks; (3) exploring the significance of studying the ecological measurement of TC, AHN, and AP in soil of ecologically sensitive areas.

## RESEARCH AREA

This research area is located in the Hailar River Basin of Inner Mongolia Autonomous Region, see Fig. 1 with a geographical location of 47°34′–0°16′N and 117°45′–122°28′E. Basin area $6.13 \times 104$ km$^2$, with a maximum width of 275 km from north to south and a maximum length of 325 km from east to west. It is connected to the New Balhu Right Banner and Manzhouli City to the west, bordered by the Great Khingan Mountains to the east by the Oroqen Autonomous Banner and Arong Banner, bordered by Erguna City and Genhe City to the north, and bordered by Zhalantun City and Arshan City in Xing'an League to the south. The watershed is located in a high latitude region, and half of the year is under the control of the cold winter monsoon, resulting in a relatively cold climate. The annual average temperature decreases from west to east, with an annual average temperature of $-1.3$ °C to $-2.6$ °C. The annual precipitation is unevenly distributed in the region, increasing from downstream to upstream. The average annual precipitation of the Hailar River over the years is between 275 and 387 mm. There are many mountainous areas, up to 531 mm. The precipitation from June to September during the flood season accounts for 80% to 84% of the annual precipitation, especially in July and August, which account for 52% to 57% of the annual precipitation.

There are currently 13 natural reserves at all levels in the Hailar River Basin. Affected by the topography of the Hulunbuir Grassland, the banks of the Hailar River are wide and flat, with a width of approximately 50 to 300 m. These broad riparian zones are rich in water, food, and biodiversity, making them ideal habitats for migratory birds and important ecological protection corridors in the region.

## MATERIALS AND METHODS

### Experimental design and soil sampling

Four typical sections were selected in the main stream of the Hailar River, the detailed spatial distribution positions of sampling points are shown in Fig. 1, namely the Yakeshi section, the downstream section of the Yimin River section, the downstream section of the Moleger River section, and the Erka Wetland section. One sampling point is set on both sides of the river (concave and convex banks) at each cross-section, with sampling points located at a distance of 1m from the river bank. The collected points are evenly distributed on both sides of the coast and are located in the flooded area of the river bank. This area belongs to the area covered by floodplain wetlands, with high ecological sensitivity and

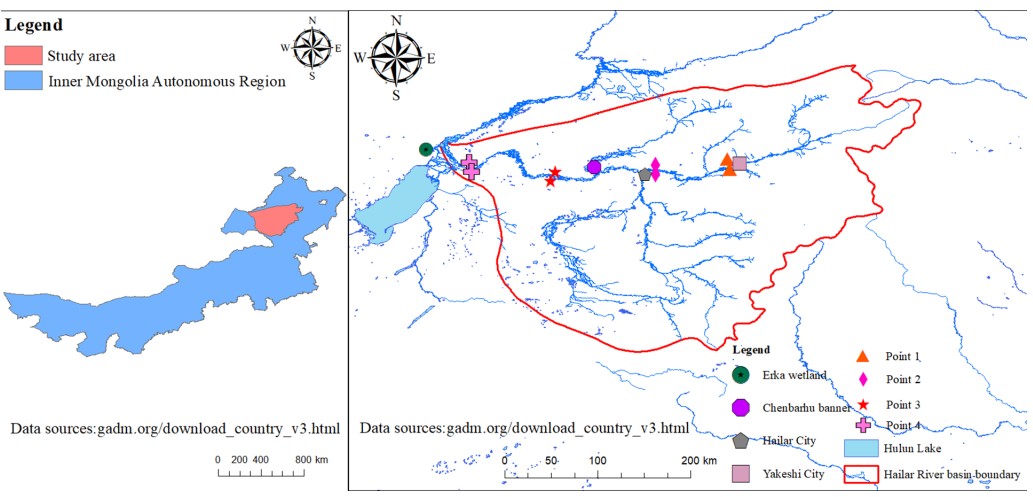

**Figure 1** **Location of the study area.** Base map and data source credit: https://gadm.org/download_country_v3.html.

is easily affected by river runoff and inundation status. For future in-depth research, it is planned to conduct repeated sampling at the same location every June to September.

Use a soil drill to collect soil at a depth of 0–20 cm and layer it in layers of 1 cm. And layer by layer according to a thickness of 1 cm. During the sampling period, pay attention to the layering of soil samples. Store self sealing bags separately for every 1 cm of soil to ensure the quality of sampling and not be affected by other sample soil layers. Due to geological conditions, some sampling points have a collection depth of less than 20 cm; The sampling schematic diagram is shown in Fig. 2. Five sample columns were collected using the plum blossom distribution method at each sampling point, and the samples from each layer of the five sample columns were evenly mixed; the vegetation coverage of the sampling points is mainly grassland, and individual points are covered with shrubbery.

The location of the sampling points is shown in Fig. 1, and the basic information of the sampling points is shown in Table 1. The sampling points are all selected from areas that have not been disturbed by human activities.

## Determination of soil physical and chemical properties

Layer the soil samples collected from the field into self sealing bags and bring them back to the laboratory for analysis. TC, AHN, and AP were selected as typical nutrients for C, N, and P in the soil. Measure the moisture content of the sample using the soil weighing method; measure particle size distribution using a laser particle size analyzer.

The content of TC (g/kg$^{-1}$) was determined using the potassium dichromate sulfuric acid oxidation method (*Li et al., 2020*). AHN (g kg$^{-1}$) was determined using the Kjeldahl method. AP (mg/kg$^{-1}$) was determined using 0.5 mol L$^{-1}$ NaHCO$_3$ extraction molybdenum antimony colorimetric method. The soil pH was measured using an Orion 720A pH meter in water at a ratio of 1:2.5 (mass/volume) for potential measurement.

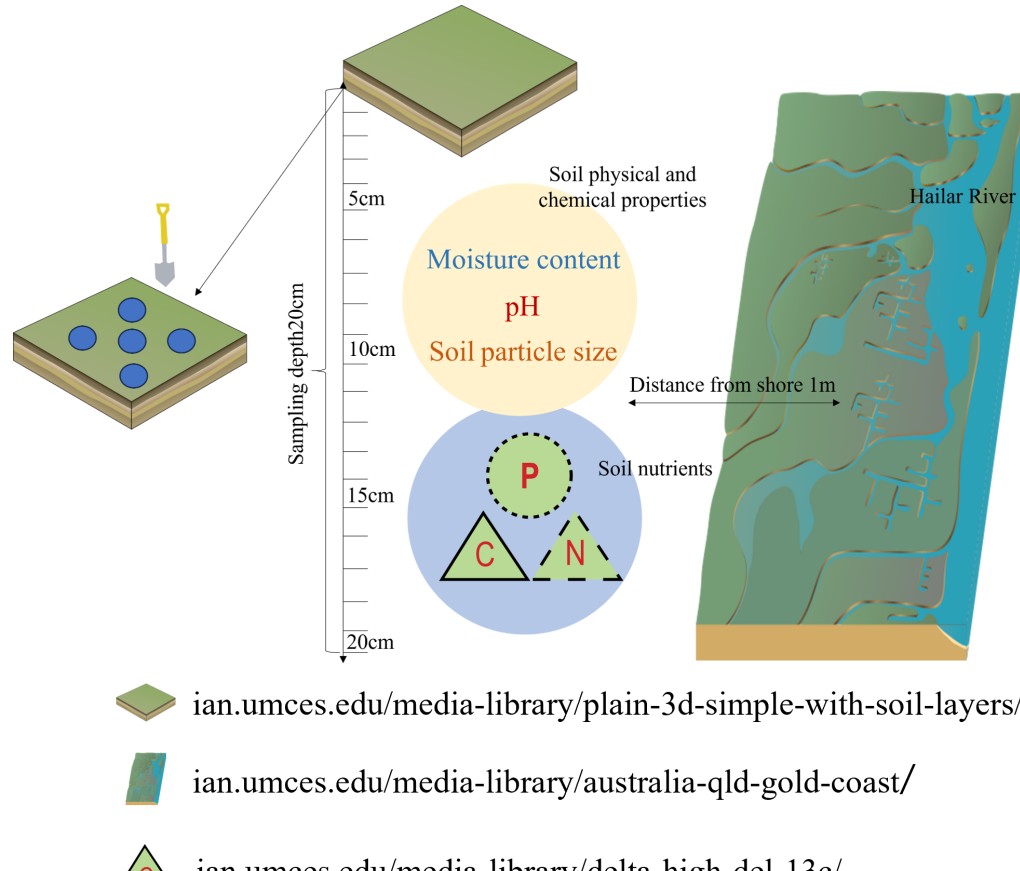

ian.umces.edu/media-library/plain-3d-simple-with-soil-layers/

ian.umces.edu/media-library/australia-qld-gold-coast/

ian.umces.edu/media-library/delta-high-del-13c/

ian.umces.edu/media-library/delta-intermittent-del-15n/

ian.umces.edu/media-library/concentration-low-phosphorus/

ian.umces.edu/media-library/spade/

Other elements from Power Point

**Figure 2 Sample collection schematic diagram.** Illustration of simple plains base with soil layers, high del 13C delta, intermittent del 15N delta, low phosphorus concentration and spade source credit: Tracey Saxby, Integration and Application Network (https://ian.umces.edu/media-library/). Illustration of Gold Coast in Queensland, Australia source credit: Kate Moore, Moreton Bay Waterways and Catchments Partnership (https://ian.umces.edu/media-library/).

## Statistical analysis

In order to determine the ecological measurements of soil TC, AHN, and AP, we selected soil particle size, water content, and soil pH from the physical and chemical properties of the soil, and explored their relationships. The concave and convex banks of the river bank were used as limiting factors for differentiation.

**Table 1  Basic information of sampling points.**

| Basic information | 1–1 | 1–2 | 2–1 | 2–2 | 3–1 | 3–2 | 4–1 | 4–2 |
|---|---|---|---|---|---|---|---|---|
| Position (E/N) | 120°37′19.39″ 49°20′42.30″ | 120°37′18.50″ 49°20′49.39″ | 119°38′18.12″ 49°16′58.86″ | 119°38′24.38″ 49°16′45.84″ | 118°54′21.65″ 49°13′2.51″ | 118°54′1.61″ 49°13′3.06″ | 117°51′2.4″ 49°29′2.25″ | 117°51′22.82″ 49°28′53.73″ |
| Riverside | concave bank | convex bank | concave bank | convex bank | concave bank | convex bank | concave bank | convex bank |
| Depth | 13 cm | 21 cm | 17 cm | 20 cm | 17 cm | 21 cm | 20 cm | 21 cm |
| Plant | grass | Shrubbery | grass | grass | grass | grass | grass | grass |

In order to further clearly demonstrate the relationship between soil physical and chemical properties and soil TC, AHN, and AP ecological metrics, important physical and chemical factors affecting soil TC, AHN, and AP ecological metrics were screened through RDA. From the basic principle, RDA is a PCA analysis of the fitting value matrix of multiple multiple linear regression between the response variable matrix and the explanatory variable matrix, and is also an extension of multi response regression analysis. RDA is a sorting method that combines linear regression with principal component analysis, aiming to find a series of linear combinations of explanatory variables that can best explain the changes in the response variable matrix, that is, the impact of the environment on the sample. Therefore, RDA is a constrained sorting of the dependent variable.

In order to further demonstrate the relationship between soil physicochemical properties and soil TC, AHN, and AP ecological metrics, important physicochemical factors affecting soil TC, AHN, and AP ecological metrics were screened through RDA. Statistical analysis was conducted on these important physicochemical factors and soil TC, AHN, and AP ecological measures using SPSS software, in this study, the chosen test model is Pearson's test, which is a parameter correlation test. Statistical testing was conducted using Origin v.2021 (Origin Pro for Windows, 2015 edition; OriginLab, Northampton, MA, USA) and Canoco. Utilize Origin Prov.2021 and Arcgis 10.2 software for mapping.

# RESULTS

## Characteristics of soil physical and chemical properties

The average particle size variation of the soil at the sampling points on the concave and convex banks is shown in Fig. 3.

The average soil particle size shows differences in the upstream and downstream of the watershed: the average soil particle size of Section 1 is 31.6~192.3 $\mu$m. The average particle size of Section 2 is 21–213 $\mu$m. The average particle size of Section 3 is 21–288 $\mu$m. The average particle size of Section 4 is 42–206 $\mu$m.

The soil moisture content of the sampling points on the concave and convex banks is shown in Fig. 4.

The average soil moisture content also shows differences in the upstream and downstream of the watershed: the average soil moisture content of Section 1 is 23.72%, while the average soil moisture content of Section 4 is 11.17%.

The pH value of the soil at the sampling points on the concave and convex banks is shown in Fig. 5.

The pH value in the upper reaches of the watershed is generally low, manifested as weakly acidic soil; the pH value in downstream areas increases, manifested as weakly alkaline soil. The pH value of the soil on the concave bank is generally higher than that on the other side, due to the abundant water in the convex bank area, which plays a role in inhibiting salt content.

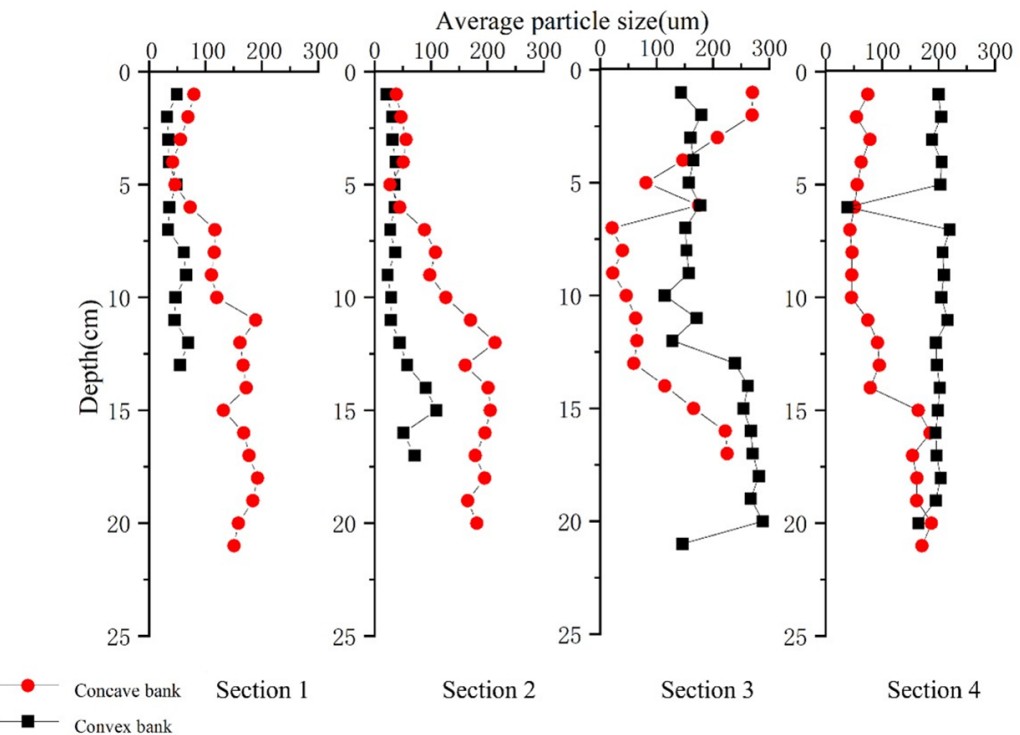

**Figure 3** Changes in average soil particle size.

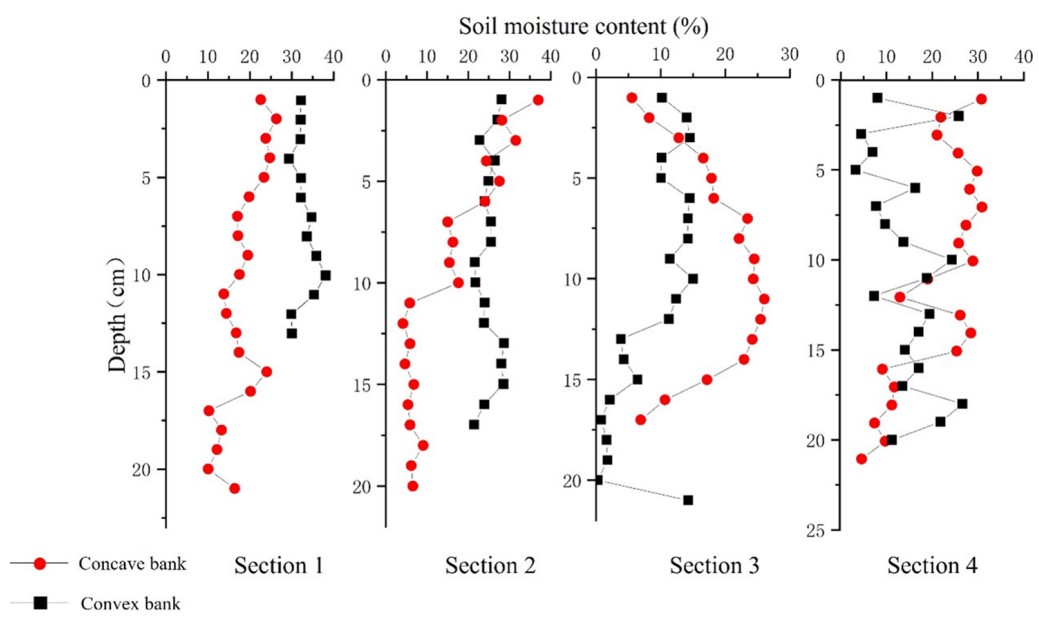

**Figure 4** Changes in average soil moisture content.

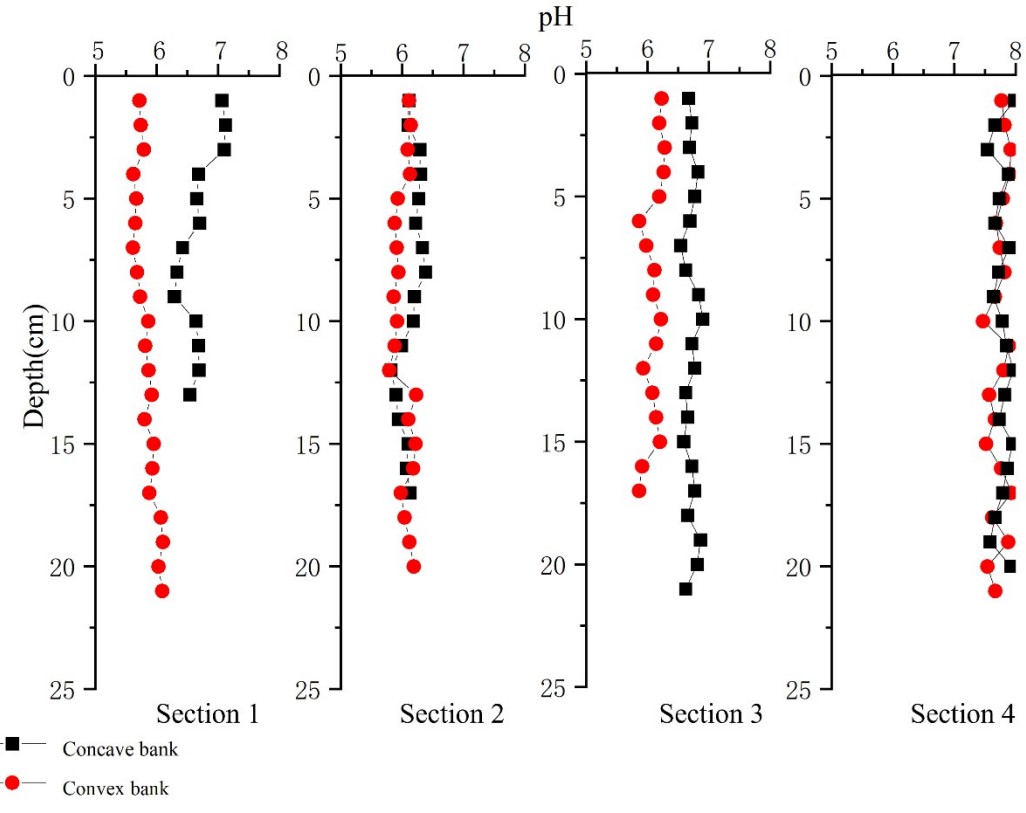

**Figure 5  Changes in average soil pH.**

## Characteristics of soil TC, AHN, and AP characteristics

The changes in soil nutrients at the sampling points on the concave and convex banks are shown in Figs. 6, 7 and 8.

There are certain patterns in the distribution characteristics of nutrient elements such as TC, AHN, and AP in the soil.

The average content of soil AHN deposition on the concave bank of Section 1 is 269.82 mg/kg, and on the convex bank is 141.53 mg/kg; The average content of concave bank in Section 2 is 119.45 mg/kg, and convex bank is 124.15 mg/kg; The average content of concave bank in Section 3 is 99.93 mg/kg, convex bank is 90.09 mg/kg, concave bank in Section 4 is 80.60 mg/kg, and convex bank is 102.52 mg/kg.

The average content of soil TC deposited on the concave bank of Section 1 is 78.00 g/kg, and on the convex bank is 21.35 g/kg; The average content of the concave bank of Section 2 is 26.18 g/kg, and the convex bank is 31.78 g/kg; The average content of concave bank in Section 3 is 8.93 g/kg, convex bank is 38.73 g/kg, concave bank in Section 4 is 6.59 g/kg, and convex bank is 28.53 g/kg.

The average content of soil AP deposited on the concave bank of Section 1 is 3.71 mg/kg, and on the convex bank is 2.99 mg/kg; The average content of concave bank in Section 2 is 12.40 mg/kg, and convex bank is 13.36 mg/kg; The average content of concave bank
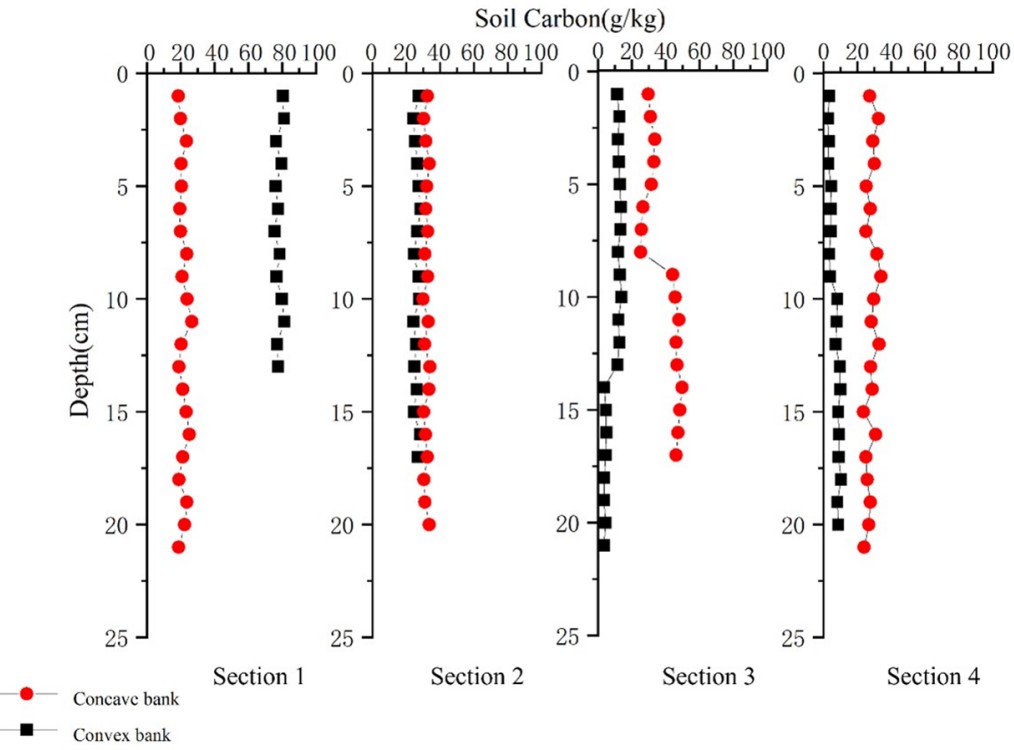

Figure 6   Changes in soil TC.

in Section 3 is 8.18 mg/kg, convex bank is 4.80 mg/kg, concave bank in Section 4 is 7.60 mg/kg, and convex bank is 7.85 mg/kg.

The potential reason for the higher content of TC and ANH in the convex bank of Section 1 compared to the concave bank is that the soil TC and ANH mainly come from the input of organic matter such as plant litter, root exudates, and microbial residues (*Zhu et al., 2022*). This may be due to the fact that Section 1 is relatively close to the Great Xing'an Mountains, and the formation of the convex bank is mainly caused by erosion deposition. Many of the deposited particles come from the upstream forest soil, which has a high organic matter content; However, Section 2 is gradually farther away from the upstream forest area, resulting in consistent TC and ANH contents in the concave and convex banks. However, downstream, due to the gradual deposition of organic matter in the soil of the convex bank upstream, the TC content in the downstream is extremely low. Therefore, this distribution pattern of content from upstream to downstream may be closely related to the source of soil TC and ANH (*Zhu et al., 2022*). The soil P content is more derived from the long weathering process of rocks (*Feng, Bao & Pang, 2017*; *Vitousek et al., 2010*), which is inconsistent with previous research results based on national soil data (*Tian et al., 2010*). The specific reason for this is not yet clear. Due to the influence of the Hailar River water outflow in the study area (*Dong & Hu, 2021*; *Dong & Hu, 2022*), the submerged state has a significant impact on the formation of river banks. Therefore, this study suggests that the occurrence of this situation may be influenced by the inundation state of the riverbank.

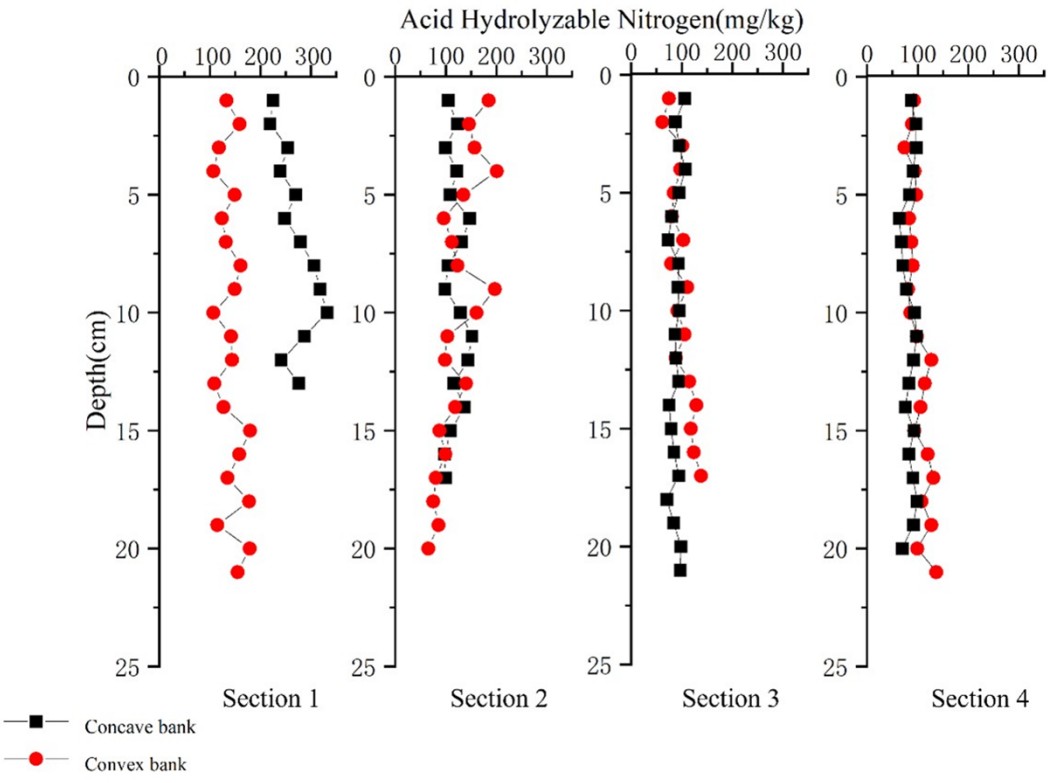

**Figure 7   Changes in soil AHN.**

## RDA analysis of soil TC, AHN, and AP ecometrics

In order to further explore the relationship between factors, RDA analysis was conducted in the ecological software Canoco 5. RDA analysis can reflect samples and environmental factors on the same two-dimensional sorting chart, from which the relationship between sample distribution and environmental factors can be intuitively seen (*Zhao et al., 2023*). This method is not limited to linear analysis of the relationship between independent and dependent variables, but rather uses multidimensional gradients to analyze the regression relationship between response variables and explanatory variables. This method has been widely applied in community research and has now expanded to more fields.

The results of the concave bank RDA are shown in Fig. 9. The contribution of soil physicochemical factors to C, N, and P varies among the four typical sections. According to the length of the arrow, for the three physicochemical factors, soil pH is the main contributing factor, followed by soil particle size, and finally soil moisture content; the results of RDA for convex banks are shown in Fig. 10. The soil physicochemical properties of convex banks and concave banks exhibit different contributions. According to the arrow length, for the three physicochemical factors, soil moisture content is the main contribution factor, followed by soil particle size, and finally soil pH.

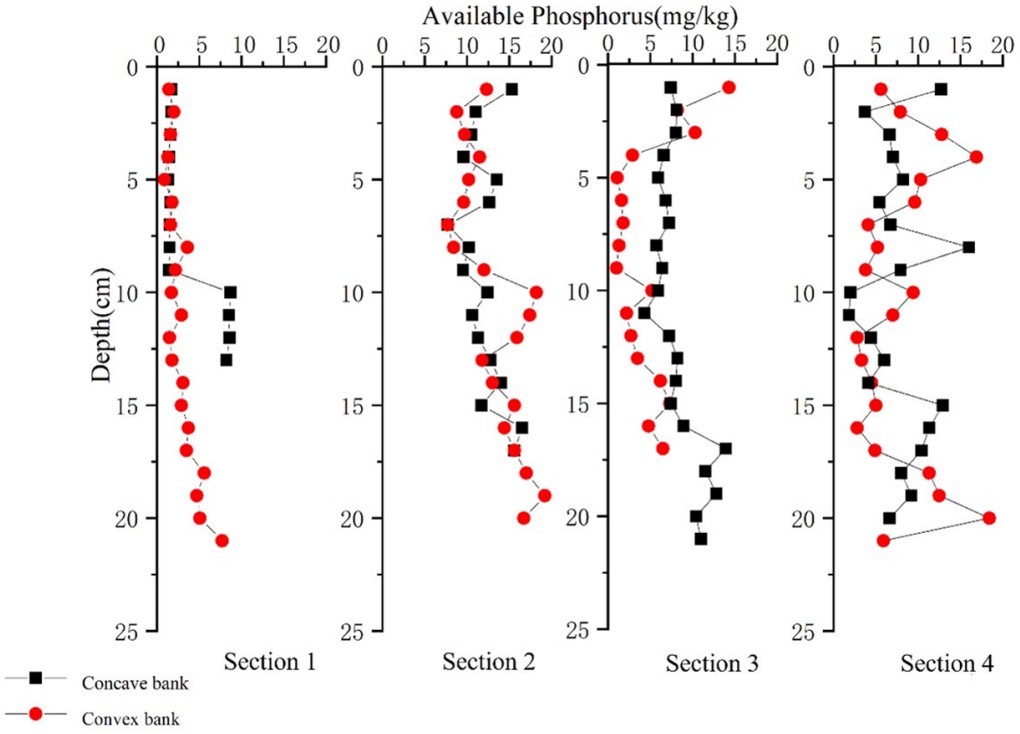

**Figure 8** Changes in soil AP.

## Relationship between soil TC, AHN, AP ecological metrology and physicochemical properties

The correlation between TC, AHN, AP ecological quantities and physicochemical properties of concave and convex bank soils in different typical sections is different, as shown in Fig. 11.

Concave bank: TC and AHN of Section 1 show a significant positive correlation with soil moisture content, while they show a significant negative correlation with soil pH; Section 2 shows a significant positive correlation between TC and soil moisture content; Section 3 shows a significant positive correlation between TC and soil moisture content, a significant positive correlation between AP and soil particle size, and a significant negative correlation between TC, AP and soil particle size; Section 4 shows a significant positive correlation between TC and soil moisture content.

Convex bank: TC of Section 1 shows a significant positive correlation with soil moisture content, a significant negative correlation with soil particle size, AP shows a significant positive correlation with soil particle size, and a significant negative correlation with soil moisture content; the AHN and TC of Section 2 show a significant positive correlation with soil moisture content, and a significant negative correlation with soil particle size, while AP shows the opposite situation; Section 3 shows a significant positive correlation between TC and soil moisture content, a significant negative correlation with soil particle size, a significant positive correlation between AP and soil particle size, and a significant negative correlation with soil moisture content; Section 4 shows a significant positive correlation

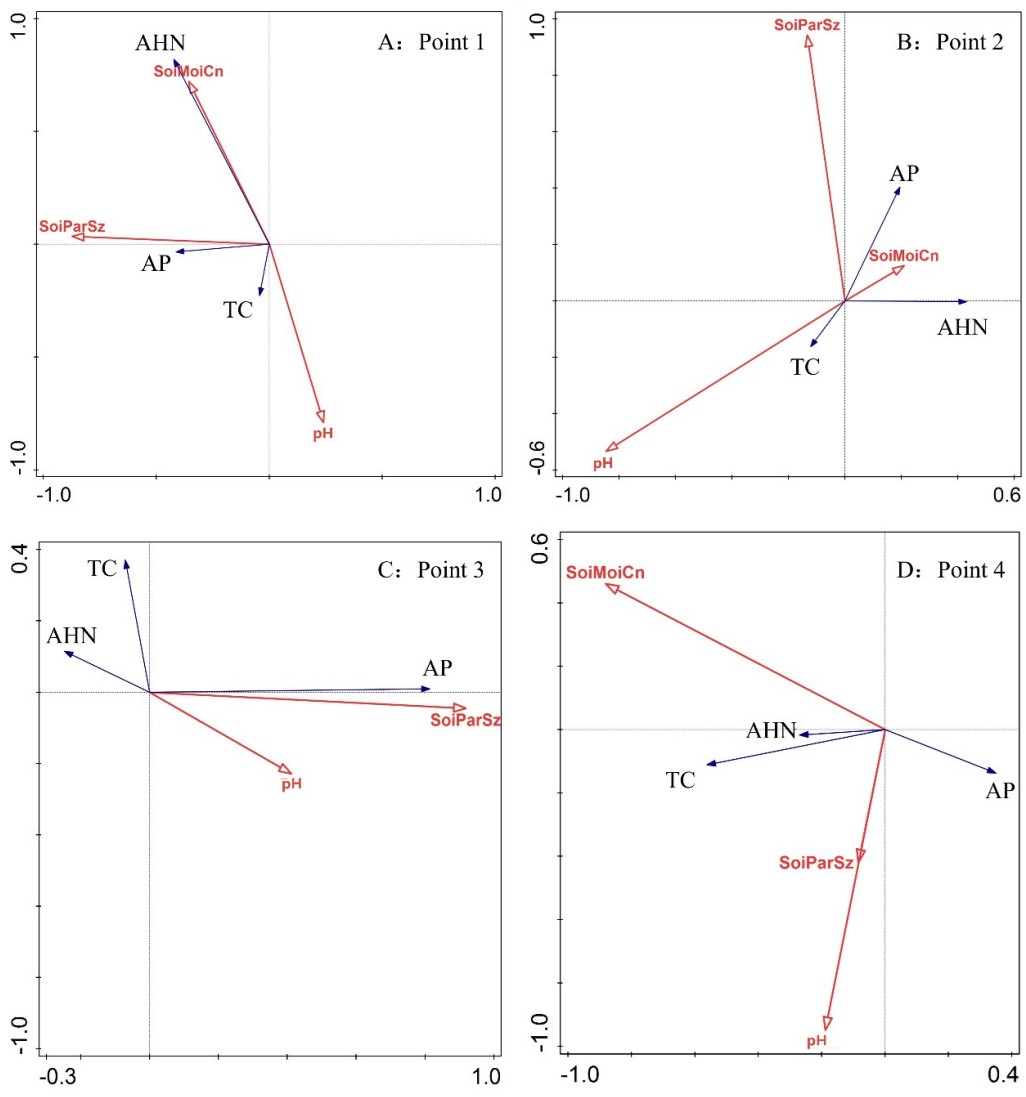

**Figure 9** RDA analysis of nutrients and physicochemical properties of concave riverbanks.

between TC and soil moisture content, a significant negative correlation with soil particle size, a significant positive correlation between AHN and soil particle size, and a significant negative correlation with soil moisture content.

## DISCUSSION

### The impact of soil physicochemical properties on the ecological measurement of TC, AHN, and AP

From the perspective of the development process of wetlands, the main pathway for nutrients to enter the wetland ecosystem through wetland sedimentation and development is to provide nutrients for the growth of plants within the ecosystem (*Harrison-Kirk et al.,*

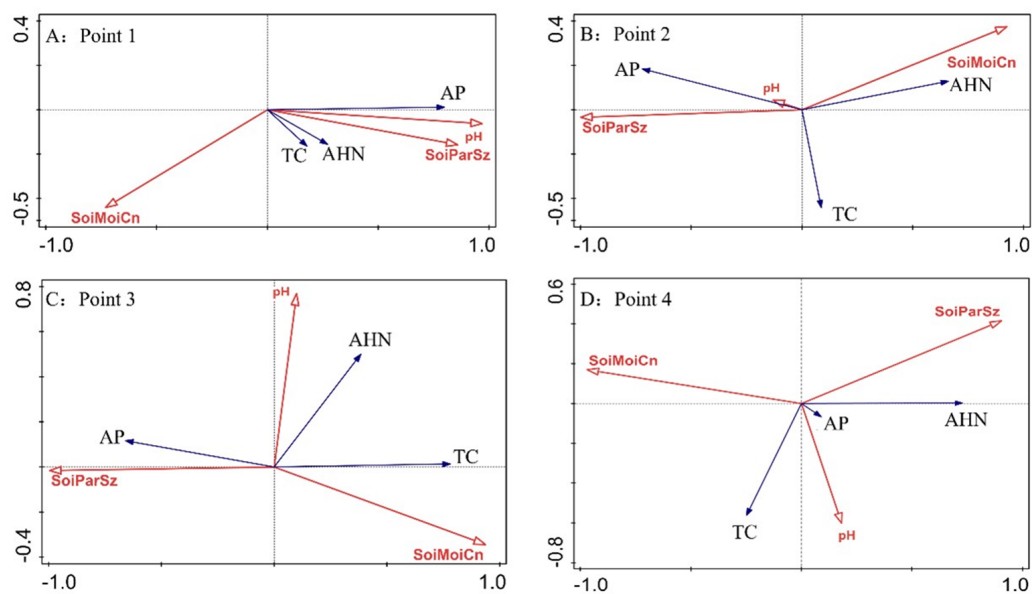

**Figure 10  RDA analysis of nutrients and physicochemical properties on convex banks.**

*2014*). Therefore, as the sedimentation process is affected, the nutrient supply within the wetland will correspondingly decrease, posing risks to ecosystem health and nutrition.

This study found through the above research that there is a significant relationship between soil physicochemical properties and nutrients and the concave and convex banks of rivers. This study found that TC, AHN, and AP exhibit regular changes in concave and convex banks at the same location, while the degree of correlation is not consistent at different locations. On convex banks, the correlation between nutrients and their physical and chemical properties, as well as with nutrients themselves, is far more complex than on concave banks. On the one hand, this may be due to convex banks being the main sedimentary area of rivers, and convex bank sediments are mainly transported and deposited by hydrology. The nutrient content of convex banks is related to sediment sources, hydrological conditions, *etc.*, thus the correlation is more complex (*Lan et al., 2022*). Long term and frequent inundation and drought conditions lead to soil organic matter content changes in litter, nutrient concentration, and texture (*Baastrup-Spohr, Møller & Sand-Jensen, 2016*; *Li et al., 2020*; *Shu et al., 2017*). On the other hand, frequent runoff pulses and hydrological dynamics on the convex bank can promote the movement of soil nutrients and plant residues from the upstream area to the downstream area (*Viparelli, Nittrouer & Parker, 2015*), resulting in a more complex correlation between various elements of the section at point 4 compared to the upstream (*Haywood, White & Cook, 2018*). At the same time, in the middle and lower reaches (Section 3) and downstream (Section 4) of the river, the water flow speed of the river is slower, and the riparian zone of these watersheds is flatter and wider than that of the upstream. Flood energy and material exchange are weaker than upstream, resulting in sediment, litter, and nutrients remaining in the downstream area (*Li et al., 2020*).

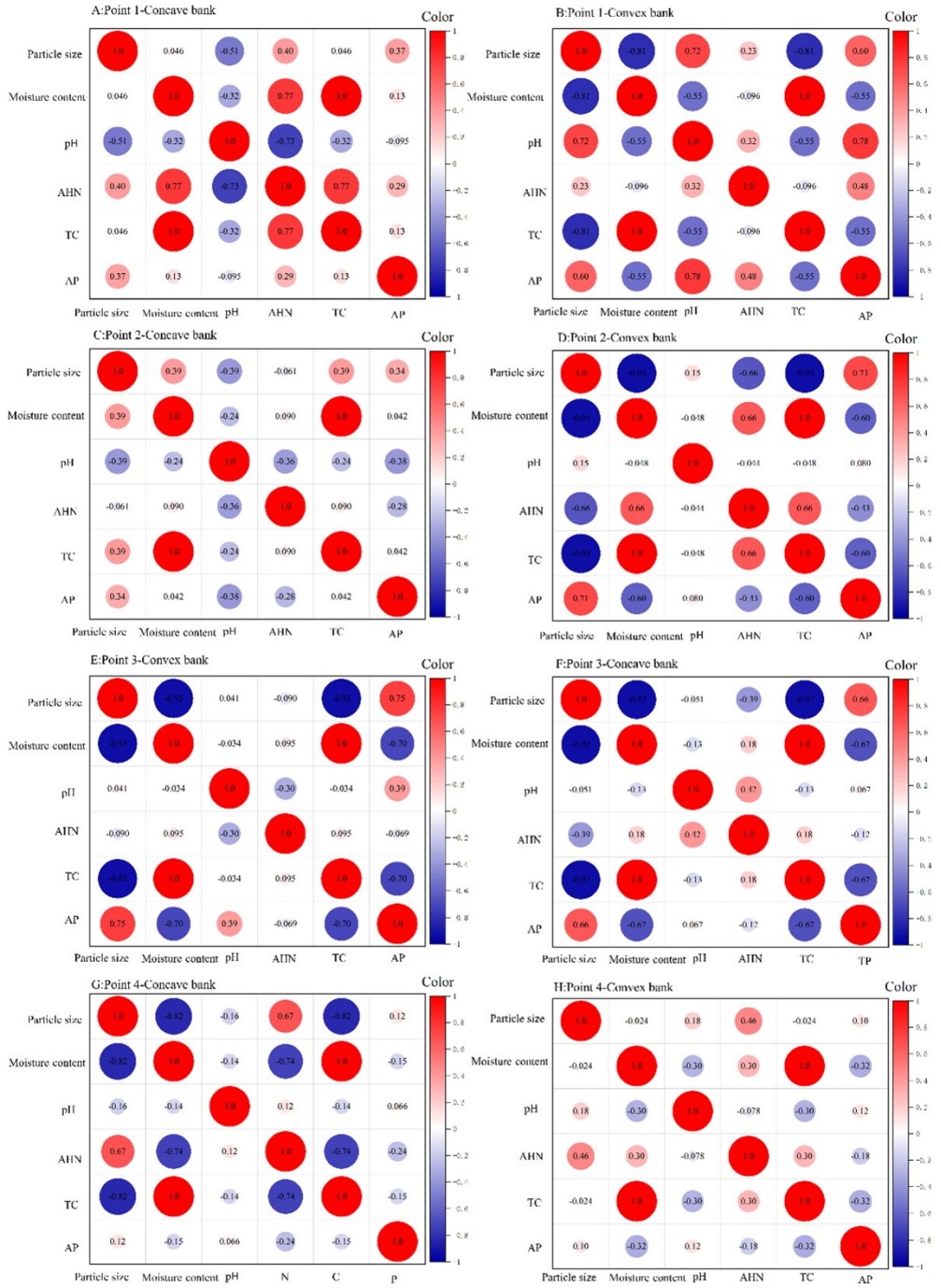

**Figure 11  Correlation between soil C: N: P ecological stoichiometry and physicochemical properties (significant correlation at the 0.05 level, with the size of the circle representing the magnitude of the correlation).** When the *p*-value approaches 1, it indicates a higher correlation between the two, and when it approaches −1, it indicates a higher negative correlation between the two.

We found that soil particle size is also an important indirect factor affecting the ecological measurement of soil TC, AHN, and AP. This may be due to the close correlation between soil hardness and soil texture, porosity, and nutrient transport, which indirectly affects the mineralization, adsorption, and ion diffusion of nutrients in the soil (*Holt et al., 2017*).

## Changes in soil TC, AHN, and AP ecometrics

In the riparian zone, the content of TC, AHN, and AP in the soil varies with the concave and convex banks of the river. When the riverbank soil is flooded with river flooding, the water flowing vertically and horizontally generates heterogeneity in soil physicochemical properties and soil nutrient formation (*Wang et al., 2018*). The heterogeneity of soil physicochemical properties can affect the cycling and ecological measurement of soil TC, AHN, and AP (*Li et al., 2020*).

This study counted the soil contents of TC, AHN, and AP on both sides of the concave and convex banks, as shown in Fig. 12. The nutrient content in the concave bank is generally higher in the upstream than in the downstream region, while the difference in nutrient content between the upstream and downstream of the convex bank is relatively small. The main reason is that the convex bank is mainly formed by the sedimentation of river particles, while the soil on the concave bank is formed by geology. The groundwater level in the downstream areas of the basin is deeply buried, and the phenomenon of salinization is relatively common. At the same time, the lower reaches of the Hailar River have a flat terrain. As the river flow decreases, the water supply to the convex bank area also decreases, and the salt inhibition effect of the surface water body weakens (*Dong & Hu, 2021*). Therefore, there are different situations of soil nutrition in the concave and convex banks, and the upstream and downstream are different. The above research also indicates that the fluctuations in the convex and straight banks are stronger than those in the concave banks, indicating that the ecological systems of the convex and straight banks are significantly affected by hydrological processes.

In addition, compared to other nutrients, the trend of changes in soil AP is significantly different from TC and AHN elements. AP is less abundant in the soil and is the main source of non renewable phosphate rocks (*Singh, Goyne & Kabrick, 2015*). The pH value of the soil on the concave bank is generally higher than that on the other side, due to the abundant water in the convex bank area, which plays a role in inhibiting salt content. The soil pH has an impact on the effectiveness of soil phosphorus. The soil pH value slows down the transportation of phosphate ions, especially inorganic phosphorus, and promotes the absorption of cations, especially divalent cations, such as $Ca^{2+}$, which further leads to the utilization and cycling of reduced P (*Jiao et al., 2016*). Sampling point 1 is located in the upstream of the Hailar River, and the main vegetation type in the Yakeshi section of the upstream area is mainly covered by trees such as Larix gmelinii in the Xing'an Mountains (*Dong & Chen, 2021*; *Guo Jinping & Zhang, 2009*). Different from the vegetation coverage dominated by grasslands in downstream areas, the decomposition rate and enrichment state of litter from trees such as larch may also have an impact on AP in soil sediments, which may be the reason why P exhibits different trends from TC and AHN changes.
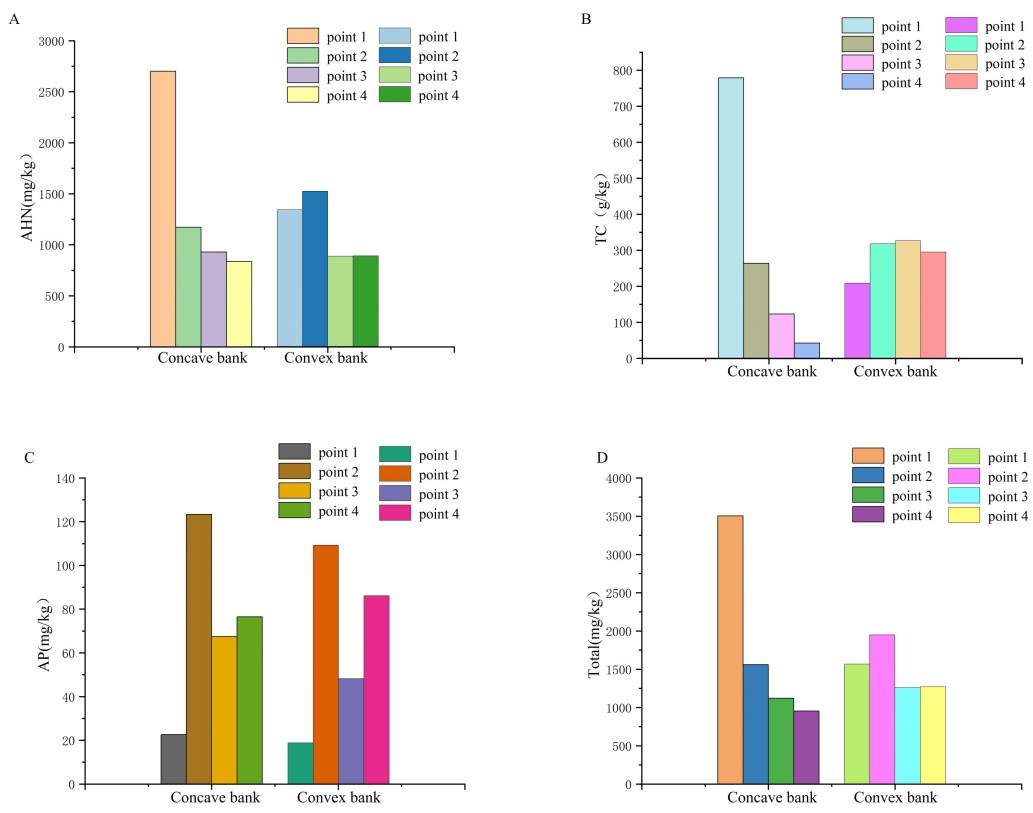

**Figure 12** **Nutrient content of concave and convex banks.**

## Potential influencing factors and research limitations

Climate change and human activities have important potential influencing factors on the ecological measurement of TC, AHN, and AP in riverbank soils, which involve multiple aspects such as soil erosion and hydrological cycling (*Abbas et al., 2023*; *Scheper et al., 2023*).

The impact of climate change on the ecological stoichiometry of TC, AHN, and AP in riverbank soil in the study area, such as changes in precipitation patterns, climate change may lead to changes in rainfall and intensity, increasing the risk of riverbank soil erosion, severe rainfall may cause mud floods, exacerbate soil erosion, and result in significant loss of riverbank soil and sediment deposition (*Torres et al., 2023*); the temperature in the Hailar region has risen, and since the beginning of the 21st century, the appearance of warm winter has led to significant ground warming. The degradation of permafrost releases groundwater to supply rivers, while the thickness of the permafrost active layer increases and seasonal freezing decreases. The changes in soil freeze-thaw have strengthened the migration of water to the underground, resulting in an increase in river runoff and a significant response to climate warming. Therefore, the temperature rise caused by climate change affects the melting of the soil permafrost layer and changes in river runoff of the Hailar River, thereby affecting the sedimentation process of the riverbank (*Dong & Hu, 2022*).

The impact of human activities on the ecological measurement of TC, AHN, and AP in riverbank soil includes: land development, construction, and pollutant emissions during urbanization and industrialization directly affect the quality and sedimentation process of riverbank soil; unreasonable agricultural practices, overgrazing, and fertilization can lead to soil erosion, increasing the risk of soil particles and nutrients being transported to riverbanks; the construction of embankments, river regulation, and renovation of water conservancy facilities may alter hydrological conditions and affect the sedimentation and erosion processes of riverbank soil (*Symmank et al., 2020*; *Kayitesi, Guzha & Mariethoz, 2022*).

There are also certain limitations in this study, such as the impact of seasonal changes on soil nutrient content, the impact of different riverbank inundation areas on soil nutrient content, and the frequency and depth of soil sample collection. These limitations will also become the focus of future research.

Overall, the potential impacts of climate change and human activities on the ecological measurement of TC, AHN, and AP in riparian soils are complex and diverse. These impacts may lead to erosion, sedimentation, texture changes, and ecosystem damage of riverbank soil. Therefore, in the protection and management of riverbank soil, it is necessary to comprehensively consider the impact of natural factors and human activities, and take corresponding measures to reduce soil erosion and protect the riverbank ecological environment.

## Exploring the significance of soil TC, AHN, and AP ecological measurements

Studying the ecological measurements of TC, AHN, and AP in typical riverbank soil can reveal the relative availability of nutrients in riverbank soil and analyze its sedimentation process (*Viparelli, Nittrouer & Parker, 2015*). The formation of riverine soil provides a good monitoring indicator for ecological areas under long-term human activities (*Studinski et al., 2012*). Due to the location of the research area in the northeast agricultural and pastoral transitional zone, which is rich in water, food, and biodiversity, it is an important ecological protection corridor in the northeast region. Therefore, based on the good correlation between monitoring changes in soil C, N, P and soil physicochemical properties, establishing a long-term monitoring database based on monitoring soil C, N, P will provide necessary data support for more effective estimation of soil nutrient content in the agricultural and pastoral transitional zone, And provide new ideas for local functional departments to reasonably manage and protect sensitive and vulnerable grazing areas along the riverbank.

With social progress, economic development, and the continuous expansion of urbanization, the impact of human activities on the global carbon, nitrogen, and phosphorus cycle is changing the content and use of these elements, resulting in potential global impacts on climate and ecosystems (*Li et al., 2020*).

Research limitations and future directions: the sampling points collected by the research area are located within the floodplain wetland. Based on the study of the floodplain wetland ecosystem, more sustained monitoring and in-depth exploration are needed to accurately evaluate its influencing factors and better protect the floodplain wetland

ecosystem (*Dong & Hu, 2022*). Ecometrics of soil C, N, and P have a profound impact on promoting nutrient cycling, ecosystem dynamics, and biogeochemical cycling mechanisms, and provide support for ecosystem management. In the future, researchers will introduce ecometrics into the study of river floodplain wetland ecosystems, focusing on the global cycling direction of C, N, and P. Soil ecometrics will serve as a bridge for further research on C, N, P cycling and their response to global climate change patterns (*Zhu et al., 2022*).

## CONCLUSION

The aim of this study is to evaluate the different impacts of soil C, N, and P ecological measures on the upper and lower reaches of the Hailar River in the Hulunbuir Grassland, as well as the main control factors. The results indicate that different sections in the upstream and downstream, as well as the concave and convex riverbanks, significantly affect soil physicochemical properties and soil C, N, and P ecological measurements.

(1) The physical and chemical properties of soil also vary in different forms of riverbanks: convex banks are the main sedimentary area of rivers, and the particle size of sediment varies during different hydrological processes, resulting in significant changes in the particle size of the soil above and below the convex banks; on the other hand, the concave bank is a eroded bank, and the particle size of the upper and lower layers of soil is relatively more uniform; the pH value in the upper reaches of the watershed is generally low, manifested as weakly acidic soil; The pH value in downstream areas increases, manifested as weakly alkaline soil. The pH value of the soil on the concave bank is generally higher than that on the other side, due to the abundant water in the convex bank area, which plays a role in inhibiting salt content.

(2) The nutrient content in the concave bank is generally higher in the upstream region than in the downstream region, while the difference in nutrient content between the upstream and downstream regions is relatively small in the convex bank. The main reason is that the convex bank is mainly formed by the sedimentation of river particles, while the soil on the concave bank is formed by geology; the difference between the upper and lower layers of TC is relatively small, while nutrients such as AHN and AP related to plant absorption and utilization exhibit fluctuations, and the fluctuations are stronger on the convex bank than on the concave bank, indicating that the ecosystem on the convex bank is significantly affected by hydrological processes

(3) The nutrient content of concave banks is often positively correlated with soil moisture content, while convex banks are positively or negatively correlated with soil moisture content and soil particle size, which further confirms our research. Convex banks are the main sedimentary area of rivers, and sediment is mainly transported and deposited by hydrology. The nutrient content is related to sediment sources, hydrological conditions, and other factors, making the correlation more complex.

Therefore, studying the content of soil C, N, and P in different riparian zones under typical cross-sections is of positive and important significance for protecting ecologically sensitive areas.

### Funding

This work was funded by the National Natural Science Foundation of China (U2102209). The funders had no role in study design, data collection and analysis, decision to publish, or preparation of the manuscript.

### Grant Disclosures

The following grant information was disclosed by the authors:
National Natural Science Foundation of China: U2102209.

### Competing Interests

The authors declare there are no competing interests.

### Author Contributions

- Xi Dong conceived and designed the experiments, performed the experiments, analyzed the data, prepared figures and/or tables, authored or reviewed drafts of the article, and approved the final draft.
- Chunming Hu performed the experiments, analyzed the data, prepared figures and/or tables, and approved the final draft.

### Data Availability

  The original data is available in the Supplementary File.

### Supplemental Information

Supplemental information for this article can be found online at http://dx.doi.org/10.7717/peerj.17745#supplemental-information.

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
