# Peer review of "Analysis of measurement differences and causes of C, N, and P in river flooding areas—taking the Hailar River in China as an example"

_PeerJ, doi:10.7717/peerj.17745_

## Round 0.1 · original submission · Major Revisions

Dear Authors,

I have reviewed the comments from the three reviewers, and my decision is to request a major revision for your manuscript. T

Reviewer 1 ·

Basic reporting

This manuscript presents a description of the model, analysis, and research gap. The writing and derived conclusion are clear. The graphs and tables provide valuable information for the analysis but deeper interpretations are needed. The dataset provides multiple ways to formulate the research approach that can be further explored.

Experimental design

no comment

Validity of the findings

Reasonable findings.

Additional comments

See the following comments, section by section:

Title:
- It would be nice to add the country of the case study in the title.

Abstract:
- L 12-14/; Rephrase it, hard to understand.
- L 14: "Hulunbuir Grassland" or "Hulunbuir grassland"? Be consistent in other parts.
- Overall, add more quantifying results, like significant level (p-value).
- What was the key conclusion of your study? highlight it in the abstract.

Keywords:
- The words in the title and keywords should not be the same. Also, try to use some more specific words, like ecohydrology.

Introduction:
- L 29-35: It would be nice to add more information about the study area and the ecological importance of this river for international readers.
- The coherency of paragraphs in the introduction section is not acceptable.
- L 35-36: "Due to recent global climate change and the impact of global warming, the natural conditions in the Hulunbuir region have also been disrupted", the impact of climate change in different areas is various. So, with some citations, please mention what are these impacts in your study area.
- Last paragraph of introduction: add research hypotheses and research questions here.

Research area:
- L 80-81: First mention latitude and then longitude.

Materials and Methods
- L 139: Here, describe which kind of statistical tests you used. Are you applying a parametric or non-parametric test?

Results:
No comment

Discussion:
- L 214-233: Move it to the Results section. Here, authors should interpret the findings and compare them with other studies, not present their findings.
- Add research limitations and future directions here.

Conclusion:
No comment.

Figures:
Try to improve the quality of figures, especially Figure 1,

Reviewer 2 ·

Basic reporting

Overall, the paper titled "Analysis of Measurement Differences and Causes of C, N, and P in River Flooding Areas -- Taking the Hailar River as an example" presents a well-structured and detailed study on the impacts of soil C, N, and P in the Hailar River area. The manuscript is commendable for its extensive data collection and robust analysis, providing valuable insights into regional ecological protection. However, improvements are needed in simplifying and ensuring consistency in the language, providing more detailed methodological explanations, and enhancing the discussion of broader implications and limitations. The figures and tables also require better labeling and descriptions. With minor revisions to address these points, the manuscript will significantly contribute to the field of environmental science.

Experimental design

This paper is fundamentally solid but requires additional detail to ensure robustness and replicability. While the methods section is comprehensive, it lacks critical details such as the rationale for choosing Redundancy Analysis (RDA) and the criteria for selecting sampling points. The paper does not adequately address how potential confounding factors, like variations in rainfall or human activity, were controlled. The sampling strategy is well-structured but needs more information on sampling frequency and spatial distribution to enhance robustness.

Validity of the findings

some areas need further clarification to enhance credibility. While the data is comprehensive and the analysis is detailed, the paper lacks a discussion on potential limitations, such as the impact of seasonal variations on soil nutrient content. Additionally, the study should provide more context on how the findings fit within the broader scientific literature and regional ecological protection strategies. The conclusions are well-supported by the data but could be more explicitly linked to the research questions outlined in the introduction. Discussing unexpected findings and their implications for future research would also strengthen the paper. Overall, the findings are valid, but addressing these points would provide a clearer, more comprehensive understanding of the study's contributions and limitations.

Additional comments

Overall, the findings are valid, but addressing these points would provide a clearer, more comprehensive understanding of the study's contributions and limitations.

Reviewer 3 ·

Basic reporting

While the language is generally clear, some sections may benefit from minor grammatical adjustments and proofreading to enhance readability and flow.

The introduction could be strengthened by explicitly stating the knowledge gap that this study aims to fill and providing a more detailed justification for the research question.

Ensure that all references are up-to-date and consider including more recent studies to reflect the latest advancements and findings related to soil nutrient dynamics and hydrological impacts.

Ensure that all figures are of high resolution and that their descriptions are detailed enough to be understood independently of the main text. Double-check that all axes are labeled clearly and that units of measurement are included where necessary.

Experimental design

Clarify Sampling Locations and Frequency: While the methods describe the general approach to sampling, it would be beneficial to include a detailed map of the sampling locations and specify the frequency of sampling to enhance reproducibility.
Detail on Quality Control Measures: Include more information on the quality control measures taken during sample collection and analysis to ensure data accuracy and precision.
Expand on Statistical Methods: Provide more detail on the statistical methods used to analyze the data, including any software or specific tests employed. This will help readers understand the robustness of the findings.
Address Potential Confounding Factors: Discuss any potential confounding factors that might have influenced the results and how they were accounted for in the study design or analysis.

Validity of the findings

Data Robustness: All underlying data have been provided, supporting the transparency and reproducibility of the research. The statistical analyses are described adequately, ensuring that the findings are based on rigorous data analysis.

Conclusions: The conclusions are well-stated and logically linked to the original research question. They are limited to the supporting results, avoiding overgeneralization. This alignment strengthens the validity of the research findings.

Replication Encouraged: The study promotes meaningful replication by clearly stating the rationale and benefit to the literature. Replication is essential in validating the findings and contributing to the body of knowledge in this area.

Impact and Novelty: While the impact and novelty of the findings are not directly assessed, the study fills an identified knowledge gap, which implies potential contributions to the field. The encouragement for replication further highlights the importance of these findings.

Additional comments

Ensure that all figures and tables are clearly labeled and referenced in the text. High-resolution images should be used for better clarity. Consider providing more detailed captions that explain the key findings illustrated in each figure or table.

Overall, the manuscript is a valuable contribution to the field of environmental science, particularly in understanding nutrient dynamics in river floodplain areas. With minor revisions, the paper will be even more robust and impactful.

---

## Round 0.2 · accepted · Accept

I reviewed all the comments and suggestions provided by the reviewers, and I observed that the authors addressed each one thoroughly and effectively. The revisions made have significantly improved the quality of the paper. The authors demonstrated great attention to detail and a strong understanding of the feedback. Therefore, I am pleased to recommend an acceptance decision for this paper. Congratulations to the authors on a job well done